# An Experimental Investigation of the Precipitation Utilization of Plants in Arid Regions

**DOI:** 10.3390/plants13050594

**Published:** 2024-02-22

**Authors:** Wei Feng, Xiaoxu Ma, Zixuan Yuan, Wei Li, Yujie Yan, Wenbin Yang

**Affiliations:** 1Department of Livestock, Xilingol Vocational College, Xilinhot 026000, China; fw350@163.com; 2Institute of Ecological Protection and Restoration, China Academy of Forestry, Beijing 100093, China; 3School of Soil and Water Conservation, Beijing Forestry University, Beijing 100083, Chinayzx0124@bjfu.edu.cn (Z.Y.); yanyujie323@163.com (Y.Y.); 4Low-Coverage Sand Control Company, Hohhot 010000, China; nmlkyywb@163.com

**Keywords:** Ulan Buh Desert, arid area, Tamarisk, reverse sap flow, atmospheric moisture

## Abstract

What represents a water source for the ecological restoration of a plant in an arid region is still up to debate. To address this issue, we conducted an in situ experiment in the Ulan Buh Desert of China, to study desert plants absorbing atmospheric water vapor. We selected Tamarisk, a common drought-salt-tolerant species in the desert, for ecological restoration as our research subject, used a newly designed lysimeter to monitor precipitation infiltration, and a sap flow system to track reverse sap flow that occurred in the shoot, branch, and stem during the precipitation event, and observed the precipitation redistribution process of the Tamarisk plot. The results showed that Tamarisk indeed directly absorbs precipitation water: when precipitation occurs, the main stem, lateral branch, and shoot all show the signs of reversed sap flow, and the reversed sap flow accounted for 21.5% of the annual sap flow in the shoot and branch, and 13.6% in the stem. The precipitation event in the desert was dominated by light precipitation events, which accounted for 81% of the annual precipitation events. It was found that light precipitation can be directly absorbed by the Tamarisk leaves, especially during nighttime or cloudy days. Even when the precipitation is absent, it was found that desert plants can still absorb water from the unsaturated atmospheric vapor; even the absorbed atmospheric water vapor was transported from the leaves to the stem, forming a reversed sap flow, as a reversed sap flow was observed when the atmospheric relative humidity reached 75%. This study indicated that the effect of light precipitation on desert plants was significant and should not be overlooked in terms of managing the ecological and hydrological systems in arid regions.

## 1. Introduction

Plants absorb water from the soil and transport it up to the leaves to participate in photosynthesis or transpiration [1,2]. This water transport process is usually described as the soil–vegetation–atmosphere continuum (SPAC) and is the primary framework for studying the transport of water through a plant. The physical basis for the SPAC is that water moves from a high water potential to low water potential [3,4]. The gradient of the water potential from the soil to the roots to the leaves drives water uptake and transport by plants. Studies have shown that leaves can also absorb water [5,6], especially Crassulaceae vegetation [7,8]; thus, water might flow in a reverse direction in the SPAC system. If the atmosphere is relatively wet with a relatively high water potential, water flows from the atmosphere through the shoot to the stem, and then to the soil.

For water from the atmosphere to enter plant leaves [9], the water potential of the leaves should be lower than the atmospheric water potential [10]. It was found that tropical montane forests are able to absorb water from the air during the dry season [11,12,13], because tropical areas are prone to the presence of rainy and foggy weather [14]. On a foggy day, water condenses and adheres to the leaf surface; this leads to a situation that the external water potential of the leaf is higher than the internal water potential of the leaf; thus, water can be drawn from the air into the leaf. Sufficient evidence has shown that a variety of pathways exist for the transport of atmospheric water into the leaf, the cuticle, the stomata [15], and the water channel protein [16]. Water transport in plants also relies on water potential gradients in different parts of the plant’s body [17]. When the water potential of the leaves is higher than the stem, water flows from the leaves to the stem. In fact, water in the leaves and stem could flow freely and replenish each other when certain parts of the plant body are dehydrated [18]. When the soil moisture is high, the water potential of the soil rises so the root system could absorb water from the soil and transfer the soil moisture to the stem and leaves for photosynthesis and transpiration [19]. Scientists have also found that the plant root system is able to transfer water to the soil system in the reverse direction when the soil layer is extremely dry [20,21].

Although researchers have made great progress in understanding absorption of atmospheric water by leaves [22], there are still several critical knowledge gaps on the subject. For example, vegetation is capable of transpiring water at night without sunlight, and the mechanism of doing so is not entirely understood [23]. Most of the current research on the subject is concerned with soil water from gas and precipitation, and the water balance formula is used to calculate the water vegetation consumption [24]. However, soil water in arid regions is mostly from condensation (liquid water) and precipitation [25,26], while atmospheric water absorbed by vegetation is less studied and the mechanism of such a process is unclear as well. A precipitation amount less than 5 mm/d will only infiltrate into the shallow soil layer whose depth is usually less than 5 cm in sandy land; thus, such precipitation events do not contribute to the soil moisture of the plant root layer (which is usually deeper than 5 cm) and is considered ineffective to desert plants [27]. However, some researchers have found that cacti could utilize water from precipitation events as small as 2.5 mm/d with succulent stems [28]. Indeed, further studies show that desert plants are more closely related to individual strong precipitation pulses rather than the total precipitation amount [29,30]. However, some other researchers have found that shrubs like Nitraria and Elaeagnus respond physiologically to light precipitation [31]. Such research mainly focuses on the utilization of light precipitation by shallow-rooted plants through the root system or the effects of precipitation pulses on the physiology (e.g., photosynthesis and transpiration) and morphology of desert plants [32], but does not address the issue direct absorption of precipitated water through the leaves.

We designed a sequence of multi-year in situ experiments in a selected arid region in Northern China to find out whether leaves could absorb precipitation water or not. Specifically, we are trying to answer the following questions: (1) If leaves can absorb water, then at what time scale and under what conditions do the plants start to absorb precipitated water? (2) Can the water absorbed from precipitation be transferred into the stem? (3) What is the exact amount of precipitated water that can be absorbed by leaves during the whole growing season? The answers to these questions can help us explain the ecological significance of precipitation events in the arid region, which are usually sporadic with highly variable intensities.

## 2. Materials and Methods

To figure out the water source for the survival of Tamarisk in the research area, we set up in-site observation experiments on two spatial scales: one was used to observe the redistribution characteristics of precipitation in the plot scale, and the other was used to observe the absorption of atmospheric water by leaves at the individual-plant scale. The soil moisture in the study site is normally less than 5%, and sometimes even below the measurement limit of the soil moisture probe (EC-5 probe measurement error range in sandy soils: ±3% volumetric water content), so it was impractical to use a soil moisture probe to monitor the soil moisture accurately. Therefore, the main observation target of this experiment was deep soil recharge (DSR), rather than the soil moisture, where DSR refers to the rate of downward soil recharge at a depth of 2 m. We used a newly designed lysimeter to monitor the amount of DSR to measure the replenishment effect of precipitation on the deep soil layer [33]. The reason to observe DSR at a depth of 2 m is that the roots of the vegetation in the study area are mainly distributed within a 0–1.5 m depth, and the maximum height of capillary rising of the sandy soil at the site is about 0.5 m, so the roots of Tamarisk will find it difficult to absorb the soil moisture at a 2 m depth. One point to note is that the deep soil moisture may replenish the shallow soil layer in the gaseous phase through upward vertical vapor flow. This factor was regarded as secondary and was not considered in this investigation. However, the vapor flow issue should not be overlooked without scrutiny and it requires specifically designed field experiments, which will be the subject of a future investigation.

### 2.1. Research Field

The research field site is located at the northeastern edge of the Ulan Buh Desert (106°00′–107°20′ E, 39°40′–41°00′ N) of China, with an average elevation of 1050 m above mean sea level (AMSL). (Figure 1A) The research area is flat and the soil type is mainly fine sand, and it has a semi-arid continental climate with an average annual precipitation of 98 mm, an average annual temperature of 6.8 Celsius, and an annual sunshine duration of 3229.9 h. The water table is approximately 9 m below the ground surface [34].

The main species of vegetation in the field site are native *Tamarisk ramosissima* (Figure 1B), domain in 80%, and the remaining vegetation species are *Haloxylon ammodendron*, *Hedysarum scoparium*, and *Caragana korshinskii*. Natural herbaceous vegetation mainly includes *Artemisia ordosica*, *Nitraria tangutorum,* etc. The experimental site is located in an artificial Tamarisk forest, with a relatively flat terrain with minor undulation [35]. The Tamarisk forest is about 30 years old, and is planted with a row space of 3 m × 2 m. The average base diameter of Tamarisk was 9.34 cm, the average height of Tamarisk was 2.95 m, and the average crown width of Tamarisk was 2.69 m × 2.32 m. After excavation, we found that the deepest root depth of native plants was about 6 m, but most of the roots of artificial vegetation were concentrated in 0–1.5 m depths. (Figure 1C)
Figure 1(**A**) The map showing the species distribution of artificial forests in northern China [36]. Tamarisk, as a shrub with low water consumption, is widely planted in the arid areas of China. (**B**) This diagram shows precipitation and condensation water hanging on Tamarisk branches in the morning in the site; (**C**) this diagram shows the observed in situ Tamarisk, where sap flow sensors are wrapped at the main stem, lateral branches, and shoot, respectively.
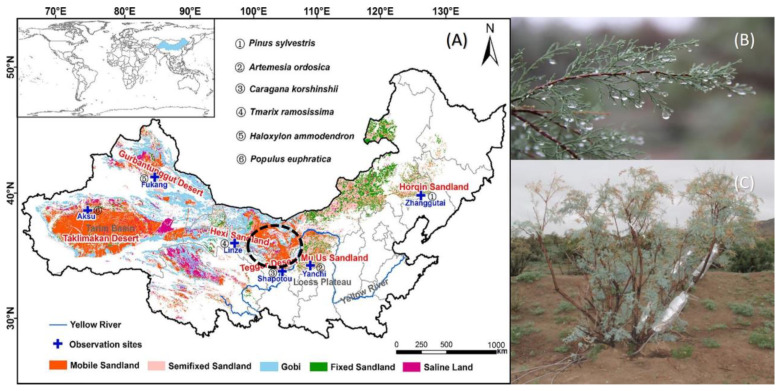


### 2.2. Sap Flow Observation

#### 2.2.1. In Situ Observation Site

The sap flow meter uses a thermal dissipation probe to measure the heat transfer rate, then converts the thermal transfer rate to the instantaneous sap flow velocity in the trunk. Long-term observation of the sap flow of plants could provide information about water exchange between plants and the atmosphere and one can use this information to monitor the impact of reforesting ecosystems on environmental changes [37]. In this research, 4 Tamarisk plants were selected as experimental plants, and the continuous sap flow data were monitored for a total duration of 150 days, covering the entire growing season of Tamarisk, roughly represented the stand structure at the site. After the experiment, both Tamarisk trees were cut down and all the leaves were collected for further analysis. We used a SF-3 HPV sap flow monitoring system (East 30, Pullman, WA, USA), which has a central heater needle and two thermistor needles up and down, and can be used in a shoot as small as 0.5 cm in diameter. To avoid thermal radiation and precipitation from interfering with data interpretation, we wrapped the probe with soft foam plastic first and then wrapped it further with tin foil plastic film. For the installation of the probe, one can refer to the SF-3 Sap Flow System Manual. The data were collected with a CR-300 (Campbell, Logan, UT, USA) at intervals of 6 min. The diameter of a Tamarisk was measured by a vernier caliper. Sap velocities were calculated according to Equation (1) [38,39]:(1)$$\begin{array}{c} V_{\mathrm{sap}}=\frac{2k}{C_{w}\left(r_{u}+r_{d}\right)}\ln\left(\frac{\Delta T_{u}}{\Delta T_{d}}\right) \end{array}$$
where *V_sap_* is the sap velocities, the unit of which is m/s; *k* is the sapwood thermal connectivity, set to 0.5 Wm^−1^K^−1^; $C_{w}$ is the special heat capacity of water; *r* is the distance of heating needle to measuring needle; ΔT is the temperature difference before and after the heating; and *u* and *d* stand for location up and down of the heater sensor.

We also needed to correct the sap velocity as the bark had been damaged during the drilling process and was affecting the sap flow results. The correction formula and sap flow calculation formulas are as follows:(2)$$\begin{array}{c} V_{c}=bV_{\mathrm{sap}}+cV_{\mathrm{sap}}^{2}+dV_{\mathrm{sap}}^{3} \end{array}$$
(3)$$\begin{array}{c} A_{\mathrm{sap}}=\pi {\left(d-d_{\mathrm{bark}}\right)}^{2}-\pi {\left(d-d_{\mathrm{bark}}-d_{\mathrm{sap}}\right)}^{2} \end{array}$$
(4)$$\begin{array}{c} F_{\mathrm{sap}}=A_{\mathrm{sap}}\times V_{c} \end{array}$$
where $V_{c}$ (ms^−1^) is the corrected $V_{\mathrm{sap}}$, and *b*, *c* and *d* are the correction coefficients; we set *b* to 1.8558, *c* to −0.0018 sm^−1^, and *d* to 0.0003 s^2^m^−2^ in line with former research [38,40]. $A_{\mathrm{sap}}$ is the sapwood area, calculated using the power law function; *d* is the measuring point diameter; $d_{\mathrm{bark}}$ is the bark thickness; and $d_{\mathrm{sap}}$ is the sapwood thickness.

#### 2.2.2. Determination of Leaf Absorption of Atmospheric Water

When the atmospheric relative humidity (RH) reached a certain level, Tamarisk leaves started to absorb atmospheric water, and such a condition was named the critical condition in this research; at this point, the reversed sap flow was monitored. As shown in Figure 2A,B, we designed an in situ control room that can simulate humidification under in situ conditions. By regulating the RH values of the control room, we can determine the critical condition when Tamarisk began to absorb atmospheric water. As shown in Figure 2B, a plastic film was laid on the lower part of the control room (ground surface) to prevent water from infiltrating into the soil layer during humidification.

### 2.3. Deep-Soil Recharge Observation

To calculate the proportion of atmospheric water (precipitation) absorbed by Tamarisk to the total water consumption in a growing season, we carried out a Tamarisk water balance observation experiment through in situ observations of precipitation, soil moisture, and DSR. To monitor DSR, a new lysimeter was used in this research [41]. This new lysimeter was assembled with two parts: a balanced part and a measurement part. As shown in Figure 2, the function of the balanced part is to ensure that the soil moisture infiltrating into this part can be completely transported to the downward measurement part. The balance part uses a cylinder with an impervious sidewall to wrap the undisturbed in situ soil column. The length of the soil column was determined based on the local soil particle size and the capillary rise (which was less than 50 cm for the fine sandy soil of the site). The advantage of this design is that when the soil at depth B (in Figure 2) reaches the saturated state, the capillary rise can at most reach depth A (see Figure 2); thus, the soil moisture cannot overflow from the top of the soil column at depth A. When there is soil moisture infiltrating into the water balance part at depth A, the soil moisture at depth B would discharge into the underneath measurement part. The amount of water discharged from the upper balance part into the lower measuring part was calculated using a rain gauge made by the American Spectrum company (Houston, TX, USA), with an accuracy of 0.2 mm. After the installation, we irrigated the sandy soil in the balance part to make sure it reached the saturation state. After this step, the excess soil moisture would be discharged from the lower boundary of the balance part to make sure the balance part maintains the saturation state. The function of the measuring part is to record the amount of water discharged from the upper balance part. The vegetation roots in the research site were mainly distributed at a depth of 0–1.5 m depth, and the upper interface of the lysimeter was installed at a depth of 200 cm to measure the DSR generated by precipitation. The installation of this new lysimeter would inevitably alter the structure of the in situ sandy soil, so we needed to install this instrument in advance, backfill the excavation with in situ soil, and allow the soil to settle for six months to one year to approximate its pre-installation status before taking the measurement data [42]. The water balance Equations (5)–(7) is shown below:(5)$$\begin{array}{c} \mathrm{ET}=P-\Delta \mathrm{SWS}-\mathrm{DSR}-R \end{array}$$
(6)$$\begin{array}{c} \Delta \mathrm{SWS}={\mathrm{SWS}}_{E}-{\mathrm{SWS}}_{B} \end{array}$$
(7)$$\begin{array}{c} \mathrm{SWS}=\sum_{i=1}^{n}D_{i}\times {\mathrm{SVC}}_{i} \end{array}$$
where *ET* is evapotranspiration, *P* is precipitation, *SWS* is soil water storage, ${\mathrm{SWS}}_{E}$ is soil water storage at the end of the year, ${\mathrm{SWS}}_{B}$ is soil water storage before the year, *DSR* is deep soil recharge, *R* = 0, and there is no runoff in the plot, $D_{i}$ is soil depth of the *i* layer, and ${\mathrm{SWC}}_{i}$ is soil volumetric water content of the *i* layer.

### 2.4. Calculation of Atmospheric Water Absorption

#### 2.4.1. Reverse Flow Measurement

The sap flow meter could measure the direction and the amount of sap flow in different scenarios, but one should be noted that there is always uncertainty or measurement bias when the sap flow meter measures the sap flow. This is because the sap flow meter only measures the sap flow through the measuring part, but there is a certain portion of immobile water stored in leaves or branches absorbed by roots or leaves, and the sap flow meter is incapable of measuring such immobile water. The reversed sap flow (water uptake) of the branches was converted into uptake per unit area using Equation (8); this amount of water absorbed per unit area could be compared to the amount of one precipitation event. As shown in Figure 2B, to accurately calculate the amount of atmospheric water absorbed by Tamarisk leaves, we used a method of repeated sampling and weighing to establish the relationship between the amount of atmospheric water absorbed and the weight of the unit leaf dry matter.
(8)$$\begin{array}{c} \mathrm{AWA}=\frac{\mathrm{SF}}{\mathrm{BCHA}} \end{array}$$
where *AWA* is the atmospheric water absorption, *SF* is the reversed sap flux, and *BCHA* is the branch crown horizontal area.

To find the critical condition for the leaves to absorb atmospheric water, we needed to continuously adjust the RH values in the control room. If the RH was kept high for a sufficiently long time, the Tamarisk leaves would continuously absorb atmospheric water. At the end of the experiment, we would cut down the whole Tamarisk, and the branches installed with the sap flow meter were picked and weighed; thus, the water retained in the leaves could be measured. A point to note was that the condensed water attached to the leaves should be removed from time to time because this portion of water could be easily mistaken for the water absorbed by the leaves. For this purpose, we used absorbent filter paper to remove the attached water on leaves to minimize the impact on the structure of the leaves. After the measurement, the leaves of the whole Tamarisk were collected and brought back to the Desert Ecohydrology Laboratory, which is located in Beijing Forestry University, for drying and weighing.

#### 2.4.2. Calculation of Leaf Water Absorption

The atmospheric water absorbed by the leaves includes the moisture stored in the leaves and absorbed by the leaves and the moisture transported downward through the sap flow after absorption by the leaves (the so-called reversed sap flow). To measure the atmospheric water absorbed by the leaves and stored in the leaves, we selected 20 Tamarisk plants at different growing stages in the experimental plot and divided them into two groups (10 plants per group): one group with humidification, and one group without humidification. From May to September 2019, we continuously monitored the sap flow for a week per month and measured the atmospheric water absorption and reverse sap flow of Tamarisk under humidified conditions. After the experiment, all the Tamarisk branches were cut off, and the Tamarisk leaves on the branches were collected and brought back to the Desert Ecohydrology Laboratory for drying. The dry matter was weighed and the water absorption of the leaves after humidification was calculated, as shown in Equations (9) and (10). Comparing the difference between the moisture content of the leaves in the humidified and non-humidified groups, one can deduce how much atmospheric water has been stored in the leaves per unit of dry mass. To convert the water absorption of Tamarisk at a single-plant scale to the water absorption of Tamarisk per unit crown size, we cut down the Tamarisk after the experiment and collected all the leaves, which were dried and weighed, and when combined with the crown width, one could calculate the absorption of atmospheric water by the Tamarisk per unit area. The main water sources in the arid region are precipitation and condensation water. Precipitation is a water source that can be consistently monitored; in this research, we focused on calculating the uptake of precipitation by leaves.
(9)$$\begin{array}{c} {\mathrm{LWC}}_{B}=\frac{W_{B}-W_{\mathrm{Dry}-B}}{W_{B}}\times 100 \end{array}$$
(10)$$\begin{array}{c} {\mathrm{LWC}}_{A}=\frac{W_{A}-W_{\mathrm{Dry}-A}}{W_{A}}\times 100 \end{array}$$
where *LWC_B_* and *LWC_A_* are the leaf water contents (in percentages) before and after precipitation, respectively; *W_B_* and *W_A_* are the leaf fresh weights (g) before and after precipitation, respectively, and *W_Dry−B_* and *W_Dry−A_* are the leaf dry weights (g) before and after precipitation, respectively.

### 2.5. Observation of Air Relative Humidity

To facilitate the computation, this research needed precipitation data and atmospheric relative humidity data from the experimental site. We established a HOBO H21 small automatic weather station on the experimental site to record temperature, precipitation, atmospheric relative humidity (RH) in and out of the control room, and other environmental information. The precipitation sensor was a rain gauge (S-RGB-M002, Meter, Pullman, WA, USA), and the air temperature and humidity sensor were S-THB-M002 (Onset, Cape Cod, MA, USA). The vapor pressure deficit (VPD) is the difference between the actual amount of moisture in the air and the maximum (saturated) amount of moisture of the air at a given site of concern. Once the air becomes saturated, water would condense to form dew or films of water over leaves. We liked to use VPD as an index to investigate the water absorption through vegetation leaves in this study. This is based on the hypothesis that a greater water stress will lead to a greater ability of the leaves to absorb water. VPD is calculated as follows:(11)$$\begin{array}{c} \mathrm{VPD}=a\times \mathrm{Exp}\left(\frac{\mathrm{bT}}{T+c}\right)\left(1-\mathrm{RH}\right) \end{array}$$
where *a*, *b*, *c* are coefficients, set as 0.611, 17.502 and 240.97, respectively [43]; *I* is atmospheric temperature at 2 m height; and *RH* is the relative humidity.

## 3. Results

### 3.1. Time for Vegetation to Absorb Atmospheric Water

As shown in Figure 3, we observed the sap flow for the main stem, lateral branch, and shoot of Tamarisk separately. In this study, when counting the sap flow amount, the lateral branch and shoot were counted as the branch. We found that the day and night sap flow rates of Tamarisk in the in situ condition varied significantly, showing a decrease in sap flow rate at night, which may be due to the lower transpiration at night. We also found that the sap flow did not converge to zero until midnight, indicating that even when photosynthesis ceases at night, Tamarisk can still carry out physiological activities and continue to absorb soil water for transpiration. After midnight, the reversal sap flow started in the shoots first, and some moments later, the reversed sap flow was also observed in the branch and main stem. On the fourth day of the observation period, as shown in Figure 3, when precipitation occurred at the night, the main stem, lateral branch, and shoot all showed signs of reversed sap flow, meaning that precipitation was transported from the leaves to the main stem. (Figure 4)

Through the above-mentioned in situ experimental observations, we found that the Tamarisk was able to absorb precipitation moisture, especially at night, and reversed sap flows could be formed at the shoot, branch, and stem. To accurately obtain the critical condition of precipitation moisture absorption by Tamarisk, we carried out a control experiment to find out at what point the Tamarisk leaves could absorb atmospheric moisture. To serve this purpose, we used RH as an indicator to identify the critical condition of the Tamarisk absorption of atmospheric water vapor under different RH conditions. The Tamarisk was enclosed in an in situ controlled-climate room, as shown in Figure 2B, isolating the possibility of the Tamarisk leaves absorbing water vapor from the atmosphere, and avoiding vapor water entering the soil during the humidification process. We started the experiment at night and started the water vapor input in the controlled-climate room to increase the RH values gradually in the controlled-climate room. As shown in Figure 5, the rate of sap flow decreased as the RH increased in the controlled-climate room, and when the RH reached 75%, the shoot began to show reversed sap flow, meaning that Tamarisk began to absorb vapor. At the same time, dew had not yet appeared in the controlled-climate room, indicating that Tamarisk was able to directly absorb unsaturated vapor moisture. As the RH value increased further and reached 90%, dew began to appear in the controlled-climate room, and we then terminated the humidification process. Afterward, the dehumidification process started in the controlled-climate room, and when the RH value of the controlled-climate room dropped to 63%, the reversed sap flow of Tamarisk disappeared. Because dew appeared at an RH of 90%, and some branches were still wet even when the RH dropped to 63%, the RH value for the critical condition should have been greater than 63%. Unfortunately, the equipment we used in-site could only observe the change in the RH of the control room, and could not observe whether the internal organs of the leaf absorbed water. Thus, we determined the leaf-absorbed atmospheric vapor by the direction of the sap flow at the shoot position. Through several humidification experiments, we determined the RH at 75% was the critical point; at this point a reversed sap flow appeared on the shoot, meaning the leaf started to absorb the atmospheric moisture. One should note that the real critical point was certainly lower than the RH of 75%, because there was a distance from the leaf to the shoot sap flow probe, and the reserved sap flow needed time to transfer; when the reverse sap flow was observed, the process of leaves absorbing atmospheric water vapor had already begun. In the future, detailed and refined observations with better equipment are needed to address this issue.

When we found that Tamarisk reverses sap flow, it appeared the same at high-RH conditions. To verify this phenomenon, we humidified the other two controlled Tamarisks; one was at a low humidifying intensity (controlled RH at 75%) and the other at a high humidifying intensity (controlled RH above 90%). As shown in Figure 5B,C, reverse sap flow occurred under both conditions and the sap flow lasted for a long period, lasting for 8 and 7.5 h until we ended the humidification experiment. The sharp increase in the RH of the air at the moment of any precipitation event and the absorption of water vapor by the Tamarisk leaves and its transmission to the stem is direct proof that Tamarisk leaves can absorb water vapor from the atmosphere under certain conditions of high RH.

### 3.2. Precipitation Events That Can Cause Water Absorption in Leaves

In the site of this study, a short period of precipitation (precipitation pulse) may not create a high-enough RH value of up to 63% over the entire experimental plot, but the liquid water drop adhered to the leaves may create a locally high-enough RH within a small area (such as the leaf scale). Now the question is as follows: is this sufficient to create a reversed sap flow? To answer this question, we observed the variation in sap flow at two different precipitation events (0.6 mm for 20 min for a short-duration precipitation event and 12 mm for 3 h for a long-duration precipitation event). It was interesting to observe that not only the long-duration precipitation event initiated the reverse sap flow, the short-duration precipitation also did, as shown in Figure 6. When the light precipitation (0.6 mm for 20 min) occurred, the atmosphere RH was as low as 17–30%; the reversed sap flow had been observed in the shoot but not in the branch and stem. This observation indicates that during a light precipitation event, atmospheric RH does not need to reach 75%; precipitation can be absorbed and stored in the leaves without being transported to the stem. When the precipitation lasted for a longer time, as shown in Figure 6B, we found that the RH rose rapidly, reaching 95%, and the reversed sap flows were observed in the stem, branch, and shoot. This shows that both light and heavy precipitation can create favorable conditions for the leaves to absorb precipitation moisture. Light precipitation in the arid region accounts for most of the annual precipitation (81%), and plants might rely on such high-frequency light precipitation events to absorb moisture through leaves.

### 3.3. Environmental Factors Affect Vapor Absorption

The sap flow rate was much higher during the daytime than at nighttime at a given VPD value. The decrease in the sap flow rate in August 2019 (Figure 7B) compared to the sap flow rate in July 2019 (Figure 7A) indicates that the physiological activity of Tamarisk started to diminish in August. As shown in Table 1, the average daytime sap flow rate of Tamarisk in August 2019 had a low Pearson correlation with VPD (R^2^ = 0.501, *p* < 0.001) (Figure 6A), and the nighttime sap flow rate was better correlated with VPD during the same duration (R^2^ = 0.718, *p* < 0.001), where R^2^ is the coefficient of determination and *p* is the *p*-value used in the Pearson correlation analysis. The correlation coefficient R^2^ was between 0 and 1. The closer the R^2^ value to 1, the closer the linear regression prediction to true value, and the more representative the fitted formula. And in general, a *p* value less than 0.05 is a significant correlation, and a *p* value less than 0.01 is a very significant correlation. In July and August of 2019, the Tamarisk sap flow rates and VPD for daytime and nighttime were best fitted with two quadratically polynomial functions. We found that Tamarisk absorbed a greater proportion of precipitation in July than in August, when Tamarisk is budding, but the relatively low precipitation and dry soil in July forced the Tamarisk leaves to take up more water from the atmosphere in order to maintain physiological activity. In contrast, August has relatively abundant precipitation and high soil moisture, so Tamarisk leaves do not need to absorb more water from the atmosphere because the Tamarisk root system can absorb water from the relatively moist soil to satisfy its physiological activities. We infer that Tamarisk absorbs water directly from precipitation as a counter-behavior to extreme drought.

The absorption capacities of leaves at daytime and nighttime were also quite different from each other. According to Table 1, the absorption capacity of Tamarisk at nighttime was relatively high. This is probably because the photosynthesis of Tamarisk ceases at nighttime, leading to less transpiration at nighttime. With the decrease in temperature at nighttime, the RH tended to reach a higher level, even saturated and dewy. Our results showed that when the RH reached 75%, the leaves could absorb unsaturated atmospheric water directly. Sap flow is an indirect method of observing leaves absorbing unsaturated atmospheric water; better equipment with higher precision such as stain labeling and isotope tracing is needed for these measurements in the future.

### 3.4. Calculation of Precipitation Absorption and Utilization

The Tamarisk leaves were the primary means for absorbing atmospheric water. If we obtained the relationship between the water absorption and the dry mass of leaves, we could estimate the total amount of precipitation absorbed by any Tamarisk leaf. As shown in Figure 8, we collected the dry leaf mass and water absorbed by the leaves after 15 precipitation events in 2019, and found that the precipitation absorption and the dry mass of leaves were positively correlated, with a coefficient of determination (R^2^) of 0.9645. These 15 precipitation events were randomly distributed during the daytime and nighttime of July (a dry month) and August (a wet month) of 2019. The amount of precipitation absorbed by the Tamarisk leaves can be calculated by weighing the dry matter mass of the Tamarisk leaves, but this method requires the destruction of the plant. We wanted to calculate the amount of water absorbed by the Tamarisk leaves by measuring the reverse sap flow. In particular, how much precipitation would be absorbed by the Tamarisk leaves?

The reversed sap flow amount was positively correlated with precipitation events, but precipitation events did not necessarily result in a reversed sap flow in the branch of Tamarisk. The occurrence of reversed sap flow was not only influenced by the amount of precipitation but was also influenced by the duration of the precipitation event. Table 2 shows the reversed sap flow in the stem and branch of Tamarisk under different precipitation events in July and August 2019. The reversed sap flow was normalized to mm per unit based on the Tamarisk canopy area and ratio of reversed sap flow to time. One could see that even for a precipitation intensity as small as 0.5 mm/d, unsaturated atmospheric moisture could be absorbed by the Tamarisk leaves. There was a relatively strong precipitation event that occurred on 2 August 2019 with an intensity of 5.2 mm/d, but the reversed sap flow of Tamarisk was not remarkable. This was probably due to the relatively short duration of this precipitation event, and consequently the stem and branch sap flows accounted for only 0.5% and 1.1% of the precipitation amount, respectively. A light precipitation event occurred in the morning of 24 July 2019, but this precipitation event was not recorded by the in situ rain gauges as the precipitation intensity was less than the 0.2 mm, lower than the minimum measurement range of the rain gauge. Surprisingly, the Tamarisk stem and branch showed significant reversed sap flows during this precipitation period. There were two precipitation events recorded at weather stations on 18 and 22 July 2019, and the reversed sap flow was only seen in the Tamarisk shoot, while the Tamarisk branch showed no or slight reversed sap flow. This was probably because the two precipitation events occurred at midday and late afternoon, during which the evapotranspiration rates were relatively high. There was a continuous precipitation event (more than 12 h) that occurred in 27 July 2019, and consequently, significant reversed sap flows were observed at the shoot, branch, and stem. In summary, light precipitation could also lead to reversed sap flow, which was also related to the duration of precipitation and the timing of precipitation.

### 3.5. Water Absorption and Consumption Characteristics of Tamarisk

As shown in Figure 9A, the rainy season started in June and ended in September, and the mean multi-year annual precipitation was less than 100 mm [42]. As shown in Figure 9B, the sap flow rate of Tamarisk showed a rising trend from June to August, indicating that this was the fast-growing period of Tamarisk. After August, the sap flow rate decreased and Tamarisk gradually entered its hibernation period. July was the main period for Tamarisk to absorb vapor from atmosphere. As precipitation increased and Tamarisk growth slowed down, the amount of water absorbed from the atmosphere gradually decreased, as shown in Figure 9C. The timing of water consumption and timing of water vapor absorption by Tamarisk were different. The maximum water consumption period of Tamarisk was in August, while solar radiation reached its maximum in July, and transpiration was the strongest in July.

The DSR throughout the year was mostly concentrated in December–April. During the five-year experimental period, the DSR accounted for 5.77% of the precipitation over the same period. The annual precipitation of the experimental site in 2020 was 84 mm, and the DSR of the same year was 5 mm. As shown in Figure 9D, precipitation patterns and DSR at the plot revealed that precipitation was mainly distributed in June and August, while Tamarisk was still budding in June and Tamarisk physiological activity began to weaken in August; thus, the amount of atmospheric water absorbed by Tamarisk leaves was small in these two months. A larger amount of precipitation occurred in August, and soil moisture became the main source to sustain the water need of Tamarisk, as shown in Figure 9E.

In the Ulan Buh Desert, light precipitation was the main type of precipitation event. Especially at the beginning of the growing season, a small amount of precipitation led to extremely dry soil, and Tamarisk was forced to absorb water from light precipitation with leaves. When the rainy season arrived, soil moisture was relatively abundant, but Tamarisk was also approaching dormancy, with less demand for water; thus, the amounts of water obtained from both the soil and leaves dropped. To survive in a water-deficit harsh environment, Tamarisk was able to mitigate its water need by taking water from multiple means such as leaves and soil.

## 4. Discussion

### 4.1. The Timing of Absorbing Atmospheric Moisture

Whether desert-area plant leaves can actively absorb atmospheric moisture was a long-debated issue [44,45]. Some researchers believed that atmospheric water can be absorbed by leaves only after condensation [46]; for example, cloud dew can be absorbed by leaves [47,48]. The potential of unsaturated atmospheric water is low; thus, it is difficult for leaves to directly absorb water from the atmosphere [49]. In this research, we took an important step forward by controlling the RH value in the in situ controlled-climate room, maintaining it between 60% and 90%, which was at an unsaturated state. Under such experimental conditions, we found that the leaves were able absorb the unsaturated atmospheric vapor (75%). In the dehumidification experiments, we found that even with an RH as low as 30%, leaves were still able to absorb moisture from droplets at the surfaces of leaves. This absorbed water was also transported to the Tamarisk stem portion, discovered by weighing experiments and stem flow observations. This is a piece of evidence showing that desert vegetation (like Tamarisk) can absorb atmospheric water directly from the precipitation and transfer water to the stem. Whether the water transferred to the stem will continue to transfer to the soil is an open question that requires further investigation. This research showed that leaves in the arid regions had multiple means of obtaining water.

### 4.2. Characteristics of Atmospheric Water Absorption by Leaves

We demonstrated that there was no significant time lag between the appearance of reversed sap flow and the occurrence of precipitation events in shoots, indicating that Tamarisk leaves can rapidly absorb water when precipitation happens. Water absorbed by leaves can be transported downward to branches and stems. Some previous studies have found that when fog appeared without apparent precipitation events, the reversed sap flow was lagging [50]. However, when precipitation events occurred, the reversed sap flow appeared simultaneously [51]. Previous studies also found that the reversed sap flow could occur soon after leaves were wet [52]. When continuous precipitation events occurred, photosynthesis and transpiration were suppressed, and the reversed sap flow occurred in both daytime and nighttime [53]. This phenomenon also occurred in the case of fog [54], lasting more than 2 h. Longer precipitation events not only produced a significant reversed sap flow, but also yielded a high absorption ratio of precipitation by leaves [55]. Even when where is no precipitation, an RH value above 63% can still lead to reversed sap flow, which occurs mostly at nighttime. During daytime, when RH was as low as 30% or VPD was sufficiently high, the shoot may experience reversed sap flow when the leaves were wet after a light precipitation event.

### 4.3. The Fate of the Absorbed Water

Although previous studies have demonstrated the pathways by which leaves absorb precipitation, a comprehensive understanding of the water absorption mechanism was missing. Some researchers have demonstrated that the water vapor absorbed by leaves entered the leaves, but it was not clear what specific organs it entered. It was also not clear whether the absorbed water vapor was absorbed in a liquid state or a gaseous state [56]. Our research controlled the RH value at 60–90% to avoid condensate and demonstrated that unsaturated atmospheric vapor could also be absorbed by leaves, and part of the absorbed moisture was transferred to the stem. However, we were not able to confirm whether the water absorbed by leaves could be further transported downward into the root system or not.

### 4.4. The Significance of Absorbed Atmospheric Water for Plants

Leaves were able to absorb unsaturated atmospheric water vapor and precipitation water directly. Unfortunately, we could not demonstrate that the water absorbed from the atmosphere could participate directly in the physiological activities of Tamarisk. We tried to use hydrogen and oxygen isotopes to label precipitation and to see whether ^18^O would appear with photosynthesis but failed to draw any affirmative conclusions due to a few reasons. The first reason was that we could not completely isolate the labeled atmospheric water vapor from entering the soil, and the soil moisture absorbed by the vegetation was also involved in the physiological process, which could have interfered with the experimental results. The second reason was that our in situ controlled-climate room could have generated a greenhouse effect, resulting in plant death.

## 5. Conclusions

Water is the most important limiting factor for plants in an arid region, and plants often suffer from drought stress during their growth. To adapt to the arid environment, plants have developed certain ways to accommodate the harsh water-deficit environments, such as leaf degradation, a thicker cuticle, a depressed stomata, and developed horizontal root systems or deep root systems. Precipitation is the main source of water in the arid study region and has an important influence on the plant growth and physiological processes. However, it is difficult for light precipitation to infiltrate into the deep soil layer. Also due to the usual intense evapotranspiration effect in arid regions, precipitated water stays in the shallow soil only for a short period of time. In this research, we analyzed the characteristics of the precipitation patterns of the research site and investigated whether the Tamarisk leaves could directly absorb the intercepted precipitation or not. The results showed that:The precipitation in the arid region was dominated by light precipitation events (with an intensity below 0.5 mm/d). Our results show that Tamarisk leaves could absorb unsaturated water vapor and precipitation directly. During the experiment, a single nighttime light precipitation was, surprisingly, absorbed by the Tamarisk leaves at a rate of 42.6%. The Tamarisk leaves can absorb precipitation moisture, even when the precipitation intensity is less than 0.2 mm/d, especially if the precipitation event occurs late at night or early in the morning.The reverse sap flow usually appears in the shoots soon after precipitation, and then in the branch and stem in turn. The rate of reverse sap flow was not only related to the amount of precipitation, but also related to the timing and duration of precipitation. Continuous precipitation results in the escalated reversed sap flow.The reversed sap flows at the stem and branch accounted for 13.6% and 21.5% of the precipitation amount, respectively, during an observation year.

In summary, the water absorption of Tamarisk leaves is very important for Tamarisk to survive in a harsh water-deficit desert environment.

## Figures and Tables

**Figure 2 plants-13-00594-f002:**
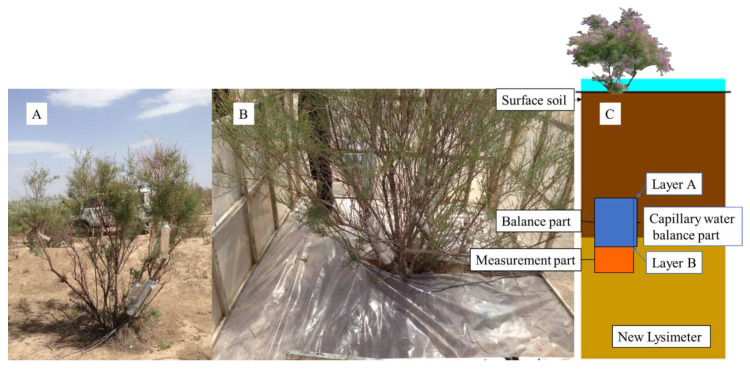
(**A**) This picture shows the in situ Tamarisk, with sap flow probes wrapped on the main stem, lateral branches, and shoots. (**B**) The figure shows the experimental setup, with the Tamarisk placed in a semi-enclosed transparent controlled-climate room, where artificial precipitation experiments can be carried out; it was closed for the atmospheric water vapor absorption experiments. (**C**) This diagram of the new lysimeter and the depth of A and B.

**Figure 3 plants-13-00594-f003:**
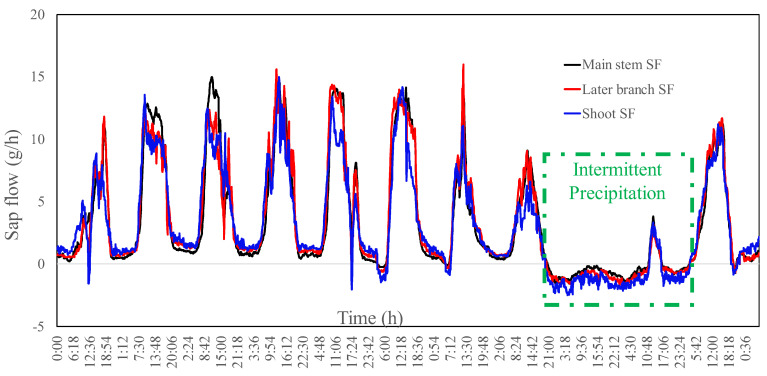
Changes in in situ Tamarisk sap flow in the main steam, lateral branch, and shoot. SF stands for sap flow.

**Figure 4 plants-13-00594-f004:**
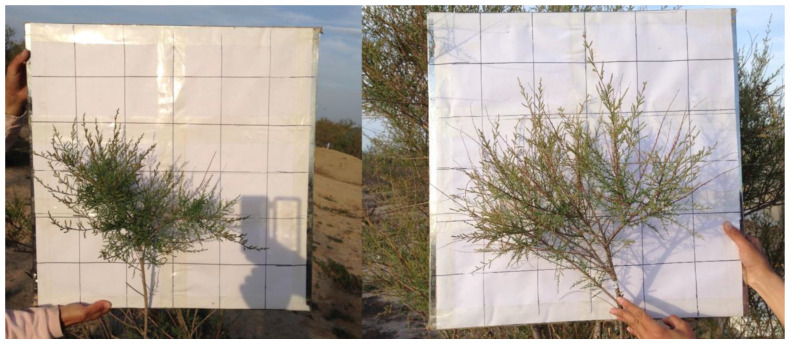
The canopy width (length, width, and height) of the target branch is measured in situ, based on the captured images.

**Figure 5 plants-13-00594-f005:**
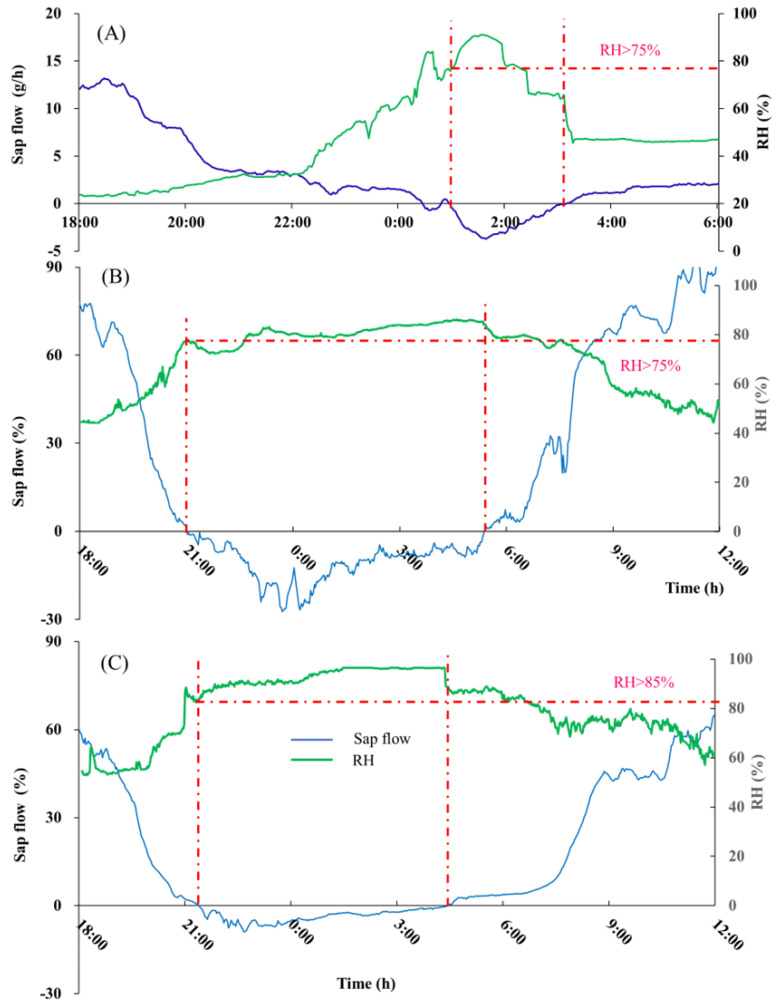
The occurrence time of reversed sap flow and corresponding RH, the humidification process and the dehumidification process. Humidifying Tamarisk 1 in the control room and finding reverse sap flow when the RH reached 75% (**A**); humidifying the Tamarisk in the other two control rooms, (**B**) is the RH around 80%, with slight fluctuations from us manually controlling the humidifier, and (**C**) is high-intensity humidification, maintaining an RH at around 90%. The purple line is RH and the red dashed line is RH reached 75% (**A**), 75% (**B**) and 85% (**C**).

**Figure 6 plants-13-00594-f006:**
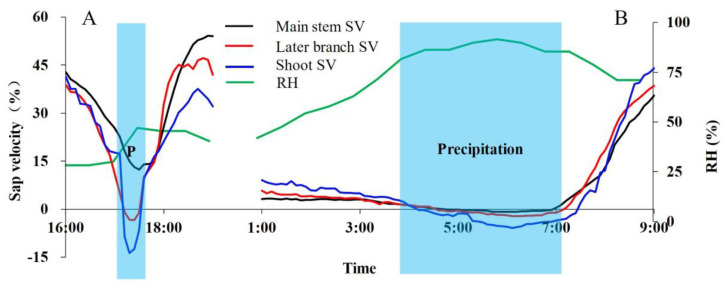
Effects of light and heavy precipitation events on reversed sap velocity (the ratio of the real-time sap velocity to the maximum sap velocity). (**A**) refers to a light precipitation during the day, (**B**) refers to a long precipitation during the night. SF stands for sap flow.

**Figure 7 plants-13-00594-f007:**
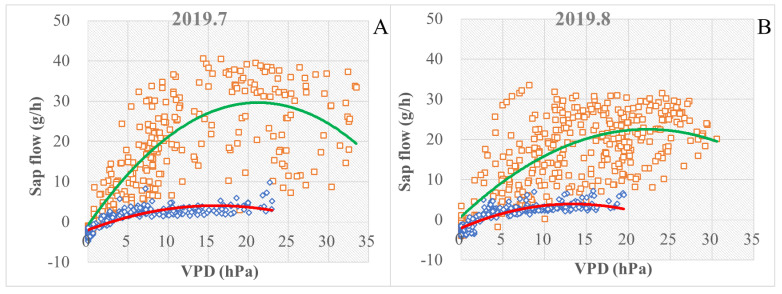
Relationship between the sap flow and water stress at different periods, where daytime (denoted as orange color □) and nighttime (denoted as blue color □) data were plotted separately. Daytime refers to the duration from 7:00 am to 7:00 pm at the same day, and nighttime refers to the duration from 7:00 pm to 7:00 am of the following day. (**A**) The mean values of branch sap flows from 18 to 28 July 2019 in relation to VPD; (**B**) the mean values of branch sap flows from 20 August to 2 September 2019 in relation to VPD. The green and red lines represent the trends of sap flow during the daytime and nighttime.

**Figure 8 plants-13-00594-f008:**
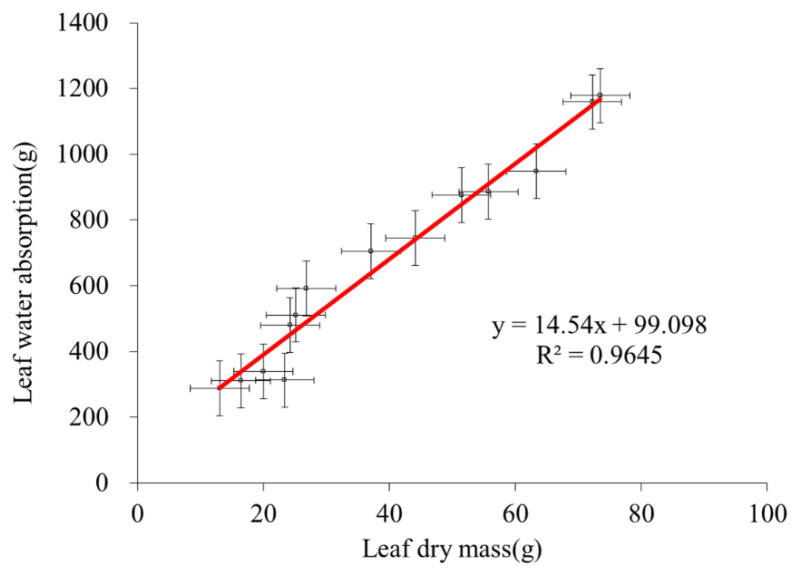
Relationship between the dry leaf mass and their absorption of atmospheric water. The red line represents the trend of relevance.

**Figure 9 plants-13-00594-f009:**
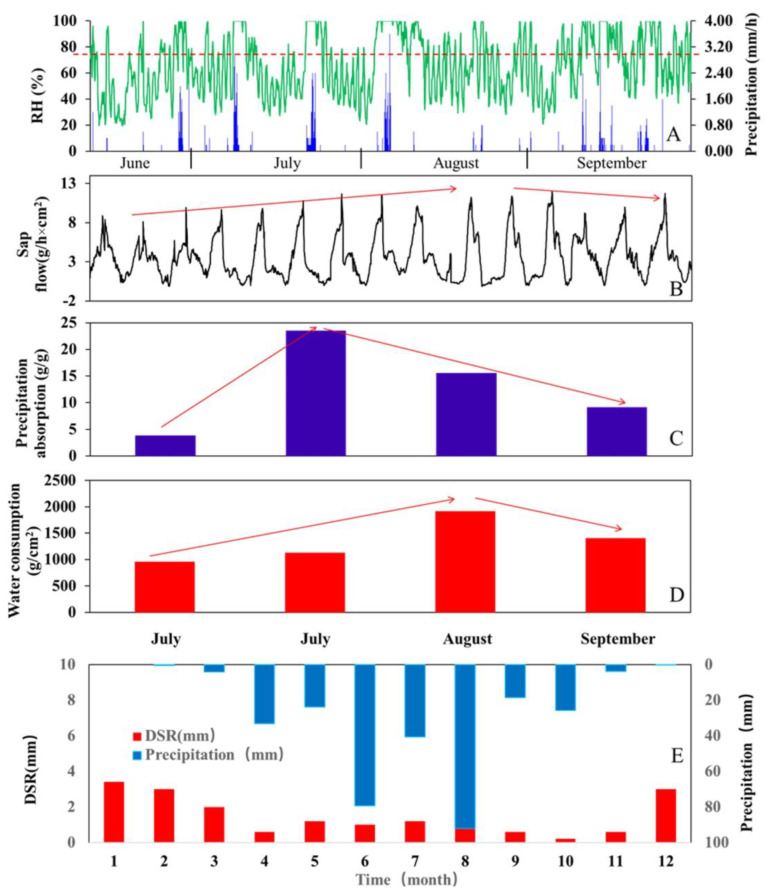
Water absorption and water consumption characteristics of Tamarisk during the growing season. (**A**) shows the relationship between RH and precipitation, the green line is RH and the blue line is precipitation. (**B**) demonstrates the sap flow. (**C**) represents the precipitation absorption of Tamarisk. (**D**) shows the water consumption during the growing season. (**E**) represents the change in DSR and the precipitation. And the red line with an arrow represents the trend in (**B**–**D**).

**Table 1 plants-13-00594-t001:** The best-fitted sap flow rates of Tamarisk branches versus VPD in July 2019 and August 2019 during both daytime and nighttime using quadratic functions. SF stands for sap flow rate (g/h) and R^2^ is the coefficient of determination.

Date	Time		R^2^
July 2019	daytime	SF = −0.044 VPD^2^ + 1.951 VPD + 0.794	0.501
	nighttime	SF = −0.033 VPD ^2^ + 0.893 VPD − 2.013	0.718
August 2019	daytime	SF = −0.068 VPD ^2^ + 2.868 VPD − 0.758	0.658
	nighttime	SF = −0.024 VPD ^2^ + 0.759 VPD − 1.981	0.703

**Table 2 plants-13-00594-t002:** Precipitation characteristics and precipitation absorption by the Tamarisk leaves during the observation period in 2019.

Date	Time	Duration (h)	Precipitation (mm)	Sap Flux (mm)		Reversed Sap Flux on Precipitation%	
				Stem	Branch	Stem	Branch
6 July	Midday	0.33	Under 0.2	0	0.004	0.000	0.020
18 July	Midday	0.5	0.2	0	0.006	0.000	0.030
20 July	Late afternoon	0.17	Under 0.2	0	0	0.000	0.000
22 July	Late afternoon	0.5	0.2	0	0.004	0.000	0.020
23 July	Afternoon	0.17	Under 0.2	0	0	0.000	0.000
24 July	Dawn	3	Under 0.2	0.002	0.01	0.010	0.050
25 July	Late afternoon	0.08	Under 0.2	0	0.006	0.000	0.030
27 July	Dawn and night	>12 h	23 (5.6; 10.4; 7.0)	5.93	9.802	0.258	0.426
2 August	Late afternoon	1.5	5.2	0.028	0.055	0.005	0.011
6 August	Afternoon	0.08	Under 0.2	0	0.004	0.000	0.020
12 August	Midnight to early morning	8	2	0.135	0.222	0.068	0.111
18 August	Afternoon	0.17	Under 0.2	0	0	0.000	0.000
25 August	Midmorning	1	0.4	0.002	0.006	0.005	0.015
27 August	Late afternoon	0.17	Under 0.2	0	0.004	0.000	0.020
30 August~1 September	Day and night	>12 h	22 (6; 7; 9)	1.312	1.608	0.060	0.073
Sum				7.409	11.731	0.136	0.215

## Data Availability

All the data are available from the corresponding author on reasonable request.

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
