# Peer review of "An Experimental Investigation of the Precipitation Utilization of Plants in Arid Regions"

_plants, 2024, doi:10.3390/plants13050594_

Round 1

Reviewer 1 Report

Comments and Suggestions for Authors

The methods of the article are sound, and the results will be of interest to the field of ecophysiology.  I've included a marked manuscript with further information about my suggestions, which are primarily focused on making sure inferences about results are moved from the Results to Discussion sections.  

There is however a more widespread issue that must be addressed before this article would be worthy of publication.  There are abundant grammatical errors that should be reviewed and corrected.  I made a few suggestions in my review, but in-depth grammatical revisions are beyond the scope of a peer review.  I suggest the authors submit the article for English proofing before a resubmission.

Comments on the Quality of English Language

My comments regarding the quality of English are detailed above.

Author Response

We thank Editor and an anonymous reviewer for their comments. Based on your comment and request, we have made extensive modification on the original manuscript. We have improved some aspects of language use. A document answering every question from the referees is also included. The manuscript has been significantly improved by addressing the comments. The following are our point-to-point responses to their comments.

Throughout this manuscript the experiment should be described in past-tense language.  I’ve made suggestions above, but the issue is too widespread to continue making grammatical corrections. Line 134.

Response: Implemented. Thanks to the reviewers' comments, we have changed the experimental procedure to describe it in the past tense in the article, and we invited native English speakers to revise the grammar throughout the manuscript.

Here and elsewhere there is future-tense wording that should be converted into past-tense wording. Line 262.

Response: Implemented. Thank you for your suggestions. We have changed the inappropriate future tense wording to the past tense wording in the article.

That information is not shown in Figure 4. Line 295

Response: Implemented. There were some errors in the file format conversion, which have been corrected. We replaced Figure 4 with Figure 3. Please see line 300.

Here and elsewhere I highlighted in yellow inferences that should be removed from the Results section and emphasized in the Discussion section instead.

Response: Implemented. Thanks to the reviewers' comments, which we fully describe in the discussion section of the manuscript. Please see line 511-516.

When this graph and a similar one elsewhere are printed in black and white the lines are indistinguishable. Consider using different patterns to the lines (dotted, solid, etc) instead of different colors.

Response: Implemented.

The wording of this statement and the two preceding ones are unclear; please reword.

Response: Implemented. We have redescribed the statement in the article. Please see line 347-350, 404-411.

Throughout the Discussion, add statements describing the results that support your inferences. Line 492.

Response: Implemented.

It would benefit your Conclusions section to link your major conclusions here to the objective questions you pose at the end of your Introduction section.

Response: Implemented. We have revised the description of the conclusion section so that it corresponds to the objective question raised at the end of the introduction.

Reviewer 2 Report

Comments and Suggestions for Authors

This paper deals with an experimental investigation about the main sources of water that feed a type of shrub plant commonly found in China desert areas. The paper is scientifically and practically sound since the subject is of great concers in periods of climate change. In fact, we still don't know where does the water that maintain different vegetation types around the world comes from. In this paper they realized a combination of in-situ measurements to test two main questions: if the plant studied (Tamarisk) is able to absorb water directly from the precipitation, 2) how is the variability of the deep water recharge at the root zone (which as considered to be around 1.5 m depth). Despite well written and designed I have some observations in relation to the presentation:

1. the summary does not reflect the entire experiment was done and the results find. It should to be improved

2. The methods could begin with the in situ area and then explain the new lysimeter functioning

3. I felt some difficulty in understand the sequence of the methods and results. As a suggestion, may be the authors should first explain the infiltration results - which will give an idea about the existence of water in the soil to be used by the plants. As a second step they could show what happen with the transpiration and the variability with precipitation.

Author Response

We thank Editor and an anonymous reviewer for their comments. Based on your comment and request, we have made extensive modification on the original manuscript. We have improved some aspects of language use. A document answering every question from the referees is also included. The manuscript has been significantly improved by addressing the comments. The following are our point-to-point responses to their comments.

This paper deals with an experimental investigation about the main sources of water that feed a type of shrub plant commonly found in China desert areas. The paper is scientifically and practically sound since the subject is of great concers in periods of climate change. In fact, we still don't know where does the water that maintain different vegetation types around the world comes from. In this paper they realized a combination of in-situ measurements to test two main questions: if the plant studied (Tamarisk) is able to absorb water directly from the precipitation, 2) how is the variability of the deep water recharge at the root zone (which as considered to be around 1.5 m depth). Despite well written and designed I have some observations in relation to the presentation:

  1. the summary does not reflect the entire experiment was done and the results find. It should to be improved

Response: Implemented. According to your suggestions, we added to findings of the maximum amount of precipitation absorbed directly by tamarisk leaves in a single experiment and the ability of tamarisk leaves to absorb precipitation moisture during microrainfall events. Please see line 567-571.

  1. The methods could begin with the in situ area and then explain the new lysimeter functioning

Response: Implemented. According to your suggestions, we adjusted the position of 2.2 and 2.3, so now the methods begin with the in-situ area and then explain the new lysimeter functioning. Please see line 134-213.

  1. I felt some difficulty in understand the sequence of the methods and results. As a suggestion, may be the authors should first explain the infiltration results - which will give an idea about the existence of water in the soil to be used by the plants. As a second step they could show what happen with the transpiration and the variability with precipitation.

Response: Implemented. Our logic was to demonstrate that tamarisk absorbs precipitation, then we counted the proportion of directly absorbed precipitation moisture to annual precipitation amount, and finally we counted the annual-scale precipitation moisture redistribution process, with a view to correctly recognising the role of tamarisk's direct absorption of precipitation moisture in the whole growing-seasonal water cycle (SPAC), and we would like to keep the whole sequence, but we have adjusted and modified any unclear parts.
